# Signaling levels mold the RAS mutation tropism of urethane

Siqi Li[†], Christopher M Counter*

Pharmacology and Cancer Biology, Duke University, Durham, United States

**Abstract** RAS genes are commonly mutated in human cancer. Despite many possible mutations, individual cancer types often have a 'tropism' towards a specific subset of RAS mutations. As driver mutations, these patterns ostensibly originate from normal cells. High oncogenic RAS activity causes oncogenic stress and different oncogenic mutations can impart different levels of activity, suggesting a relationship between oncoprotein activity and RAS mutation tropism. Here, we show that changing rare codons to common in the murine *Kras* gene to increase protein expression shifts tumors induced by the carcinogen urethane from arising from canonical $Q_{61}$ to biochemically less active $G_{12}$ *Kras* driver mutations, despite the carcinogen still being biased towards generating $Q_{61}$ mutations. Conversely, inactivating the tumor suppressor p53 to blunt oncogenic stress partially reversed this effect, restoring $Q_{61}$ mutations. One interpretation of these findings is that the RAS mutation tropism of urethane arises from selection in normal cells for specific mutations that impart a narrow window of signaling that promotes proliferation without causing oncogenic stress.

*For correspondence:
count004@mc.duke.edu

Present address: †Division of Human Biology, Fred Hutchinson Cancer Research Center, Seattle, United States

Competing interests: The authors declare that no competing interests exist.

## Introduction

The RAS genes are mutated in a fifth or more of human cancers (*Prior et al., 2020*), which is well established to be tumorigenic (*Pylayeva-Gupta et al., 2011*). There are three individual RAS genes in humans, *KRAS, NRAS,* and *HRAS*, three primary sites mutated in human cancers, $G_{12}$, $G_{13}$, and $Q_{61}$, and six possible amino acid substitutions at each site arising from a point mutation. As such, there are 54 possible oncogenic point mutations, not including rare non-canonical mutations (*Hobbs et al., 2016*). Mapping these mutations to human cancers reveals a distinct pattern or 'tropism' in which individual cancer types are characterized by specific RAS mutations (*Li et al., 2018*). The same can be said for other oncogenes. For example, the most frequent EGFR point mutation in glioblastoma is $G_{598}$V/A, but $L_{861}$Q in lung adenocarcinoma; the most frequent IDH1 mutation in low-grade glioma is $R_{132}$H, but $R_{132}$C in melanoma, and so on (*Chang et al., 2016*). While these mutational biases are well described, the mechanism responsible is not.

One hint to the mechanism underlying RAS mutation tropism is that oncogenic RAS mutations are thought to occur early. Focusing on lung cancer, in humans oncogenic *KRAS* mutations have been detected in premalignant lesions (*Kanda et al., 2012*) as well as in multiple regions within the same tumor (*Zhang et al., 2014a*), indicative of an early origin (*Wistuba and Gazdar, 2006*). In mice, oncogenic *Kras* mutations are capable of initiating tumors in carcinogen (*McCreery and Balmain, 2017*) and genetically engineered (*Kwon and Berns, 2013*) lung cancer models. As early driver mutations, it follows that RAS mutation tropism is a reflection of the normal cells in which the mutation first occurred. By its very classification, oncogenic RAS can induce proliferation (*Pylayeva-Gupta et al., 2011*). However, high oncogenic signaling through the MAPK effector pathway of RAS can paradoxically induce an oncogenic stress response in normal cells mediated by the tumor suppressors p16 and p53, which leads to the growth arrest termed senescence (*Muñoz-Espín and Serrano, 2014*). Indeed, hyperactivation of MAPK signaling via the combination of $Kras^{G12V}$, $Braf^{D631A}$, and loss of the remaining wildtype *Braf* allele activated p53 and impeded lung tumorigenesis, an effect rescued by pharmacological inhibition of the MAPK pathway (*Nieto et al., 2017*).

Accumulating evidence suggests that different oncogenic mutations can be biochemically distinct (*Muñoz-Maldonado et al., 2019*; *Smith et al., 2013*). Relevant to this study, a $G_{12}D$ mutation sterically inhibits the catalytic cleft (*Parker et al., 2018*) while $Q_{61}R$ replaces the catalytic amino acid (*Buhrman et al., 2010*). A direct comparison of $G_{12}D$ and $Q_{61}R$ mutations in Nras revealed that the former has lower GTP loading and tumorigenic potential in the skin (*Burd et al., 2014*) and hematopoietic system (*Kong et al., 2016*). Indeed, a panel of $G_{12/13}$ Kras mutants introduced by Cas9-mediated gene editing in the lung revealed widely different tumorigenic potentials between different mutants (*Winters et al., 2017*). Taken together, we hypothesize that the level of oncogenic signaling may dictate the type of mutation conducive to initiate tumorigenesis in a normal cell, and hence play a role in RAS mutation tropism.

One challenge to testing this hypothesis is trying to backtrack to catch a single, ostensibly random mutagenic event in one gene from one normal cell, decades before manifesting as cancer in humans. However, in mice the moment of tumor initiation can be precisely defined as the point of carcinogen exposure, with the added benefit that carcinogens mimic the spontaneous nature of human cancers. Carcinogenesis is also an ideal model of RAS mutation tropism. Case in point, the carcinogen urethane found in fermented foods and alcoholic products (*Gowd et al., 2018*) primarily induces pulmonary tumors with a very specific $Kras^{Q61L}$ or $Kras^{Q61R}$ driver mutation, depending on the strain (*Dwyer-Nield et al., 2010*). These oncogenic mutations are also a strong match to the mutation signature of this carcinogen. Urethane-induced mutations conform to the A>T/G consensus sequence derived from comprehensive whole-exome sequencing urethane-induced tumors (*Westcott et al., 2015*), and the even more restricted C$A$N>C$T/G$N sequence determined shortly after urethane exposure by different ultra-sensitive sequencing approaches (*Li et al., 2020*; *Valentine et al., 2020*). Such mutations at position C$A_{182}$A give rise to the $Q_{61}L/R$ oncogenic mutations characteristic of this carcinogen. We thus capitalized on this extreme bias of urethane for $Kras^{Q61L/R}$-mutant pulmonary tumors to genetically elucidate the effect of oncogenic signaling levels on RAS mutation tropism.

To explore the effect of oncogenic signaling levels on the selection of initiating oncogenic mutations, we genetically enhanced *Kras* translation to increase oncogenic activity or inactivated p53 to inhibit the cellular response to oncogenic stress in mice exposed to urethane. In regards to the first genetic change, we compared the native $Kras^{nat}$ allele that is naturally enriched in rare codons and correspondingly poorly translated (*Lampson et al., 2013*) to the $Kras^{ex3op}$ allele, in which 27 rare codons in exon 3 (which is not the site of oncogenic mutations) were converted to common, leading to roughly twice as much Kras protein in the lungs or derived cells of mice (*Pershing et al., 2015*). In regards to the second genetic change, we evaluated retaining or conditionally inactivating the *Trp53* gene specifically in the lung, which has been shown to suppress oncogenic stress due to high mutant Kras expression in this tissue in vivo (*Feldser et al., 2010*; *Junttila et al., 2010*). Using this approach, we show here that the canonical $Q_{61}L/R$ mutations are selected against in the more highly expressed $Kras^{ex3op}$ allele in urethane-induced tumors, even though the carcinogen favors this mutation. Instead, biochemically less active $G_{12}$ mutations are detected, which, upon the loss of p53, partially shifts back to $Q_{61}L/R$ mutations. p53 loss also promoted the expansion of the normally rare $G_{12}$ mutants in the $Kras^{nat}$ allele, which was accompanied by an imbalance between the mutant and wild-type transcripts suggestive of higher expression. Moreover, tumors characterized by $Q_{61}$ mutations or the less active $G_{12}$ mutants when coupled with allelic imbalance exhibited similar transcript levels of three genes known to be activated by oncogenic RAS, suggesting a similar degree of oncogenic signaling. Taken together, these data support a narrow window of signaling conducive to initiate tumorigenesis in a normal cell. Namely, that imparted by the more active $Q_{61}$ mutations in $Kras^{nat}$ allele or less active $G_{12}$ mutations in the higher expressed $Kras^{ex3op}$ allele. p53 loss reprograms this mutation tropism, allowing more active mutations $Q_{61}L/R$ in the $Kras^{ex3op}$ allele as well as promoting or permitting allelic imbalance of less active $G_{12}$ mutations in the $Kras^{nat}$ allele to now induce tumorigenesis. Selection for an oncogenic mutation imparting an optimal level of signaling in normal cells thus appears to influence the RAS mutation tropism of urethane.

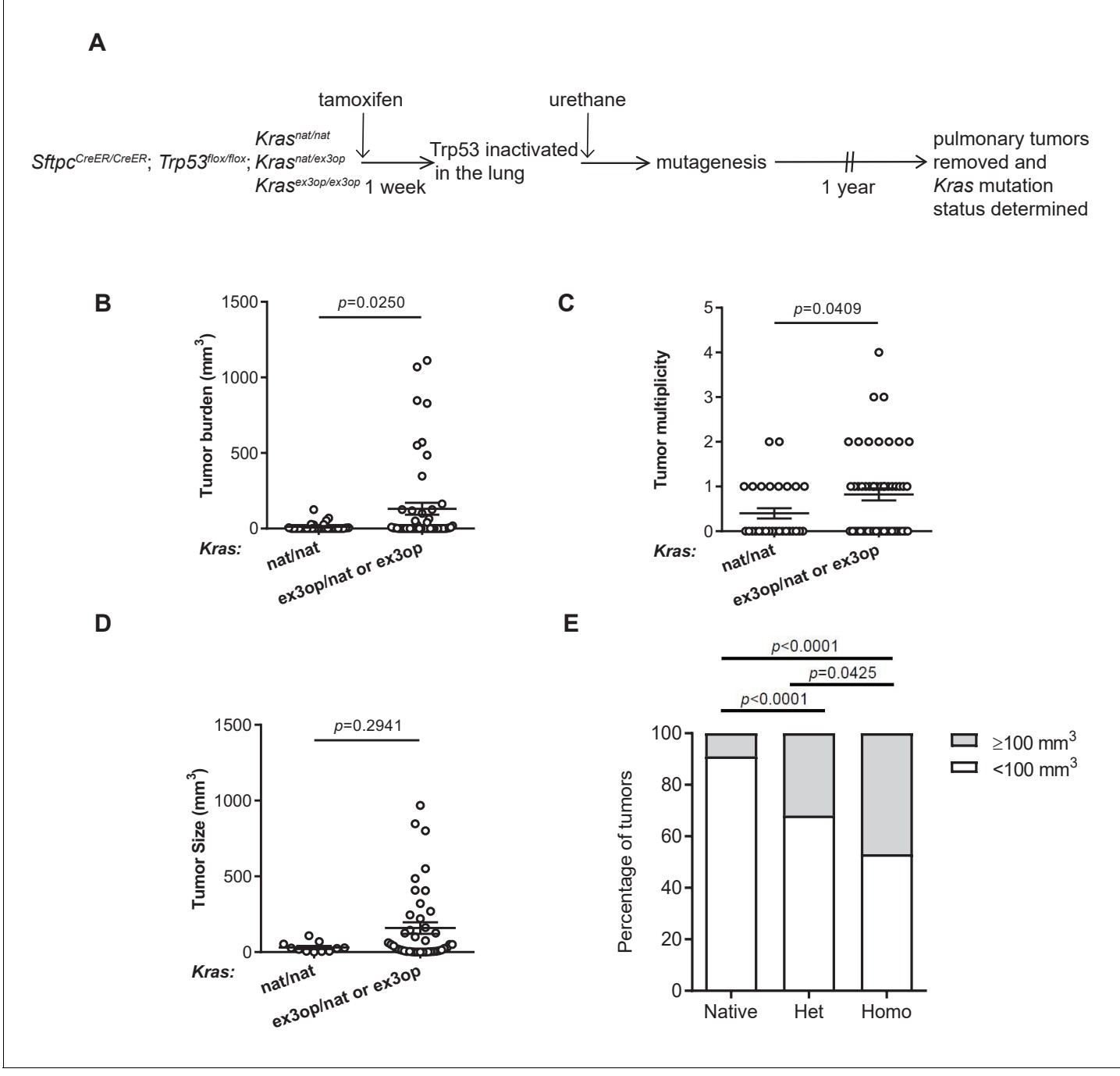

**Figure 1.** Loss of p53 converts the *Kras^ex3op* allele from suppressing to enhancing urethane carcinogenesis. (A) Experimental design to evaluate the effect of inactivating p53 specifically in the lung on urethane carcinogenesis upon increase in Kras expression. (B–D) Mean ± SEM of urethane-induced tumor (B) burden, (C) multiplicity, and (D) size in tamoxifen-treated *Sftpc^CreER/CreER*;*Trp53^fl/fl* mice in a homozygous native (B, C: n = 30 mice; D: n = 11 tumors) and heterozygous or homozygous (B, C: n = 51 mice; D: n = 42 tumors) *ex3op Kras* background. Mann–Whitney test. (E) % of tumors ≥ (gray bar) or < (white bar) 100 mm³ in tamoxifen-treated *Sftpc^CreER/CreER*;*Trp53^fl/fl* mice in *Kras^nat/nat* (n = 11 tumors), *Kras^ex3op/nat* (n = 25 tumors), or *Kras^ex3op/ex3op* (n = 17 tumors) backgrounds after urethane exposure. Two-sided Fisher's exact test.

The online version of this article includes the following source data and figure supplement(s) for figure 1:

**Source data 1.** Raw data for *Figure 1B–E*, *Figure 1—figure supplement 1B–E*.

**Figure supplement 1.** The effect of *Kras^ex3op* allele on urethane-mediated lung tumorigenesis in the absence of p53.

## Results

### The *Kras^ex3op* allele is more tumorigenic in the absence of p53

To explore the effect of oncogenic signaling levels on the selection of initiating oncogenic mutations, we genetically inactivated p53 to inhibit the cellular response to oncogenic stress or enhanced *Kras* translation to increase oncogenic activity in mice exposed to urethane. With regards to the first genetic manipulation, we crossed *Sftpc^CreER/CreER*;*Trp53^flox/flox* mice into a background with two native *Kras* alleles (*Kras^nat/nat*) or with one (*Kras^nat/ex3op*) or two (*Kras^ex3op/ex3op*) copies of the aforementioned *Kras^ex3op* allele in which rare codons were altered to common in exon 3 (*Pershing et al., 2015*). The *Sftpc^CreER/CreER*;*Trp53^flox/flox* genotype was chosen as injection of tamoxifen into such mice leads to recombination and inactivation of the endogenous *Trp53^flox* alleles in the type II alveolar cells of the lung (*Xu et al., 2012*), which is reported to suppress oncogene-induced senescence/ apoptosis induced by oncogenic Kras in this organ (*Feldser et al., 2010*; *Junttila et al., 2010*). With regards to the second genetic manipulation, the *Kras^ex3op* allele was chosen as a way to increase Kras protein expression of the native gene while leaving the rest of the locus almost entirely intact (*Pershing et al., 2015*). Cohorts of 21–30 mice from each of these three genotypes were injected with tamoxifen to inactive the *Trp53* gene in the lung, followed by exposure to urethane via a single intraperitoneal injection to induce *Kras* mutations. One year later these mice were humanely euthanized, the number and size of tumors determined at necropsy (*Figure 1A* and *Supplementary file 1*), and the tumors removed and recombination of the *Trp53^flox* alleles confirmed by PCR (*Figure 1—figure supplement 1A* and *Supplementary file 1*). As a first step, we simply examined the effect of p53 loss on tumorigenesis when Kras expression was altered. In sharp contrast to the previous findings that the *Kras^ex3op* allele *reduced* urethane carcinogenesis (*Pershing et al., 2015*), the loss of p53 instead *increased* tumor burden in mice with at least one *Kras^ex3op* allele (*Figure 1B*), similar to what was observed in a whole animal *Cdkn2a* null background (*Pershing et al., 2015*). This appeared to be a product of more tumors (*Figure 1C*), with a trend towards larger tumors (*Figure 1D*) that reached statistical significance when the data were censored for large ($\geq 100$ mm$^3$) tumors (*Figure 1E*). There was also a trend when the genotypes were subdivided into one or two *Kras^ex3op* alleles compared to the *Kras^nat/nat* background, although no difference was observed between *Kras^nat/ex3op* versus *Kras^ex3op/ex3op* genotypes (*Figure 1—figure supplement 1B–E*). This suggests that in the absence of p53 the *Kras^ex3op* allele promotes both the initiation and progression of urethane-induced lung tumors, consistent with p53 suppressing oncogene toxicity to allow oncogenic mutations in the *Kras^ex3op* allele to exert a more potent signal to drive tumorigenesis.

### Loss of p53 reprograms the extreme RAS mutation tropism of urethane

To specifically address the effect of genetically inactivating the *Trp53* gene on the type of oncogenic mutations arising in the native *Kras^nat* versus the codon-optimized *Kras^ex3op* alleles in tumors induced by urethane, we compared the *Kras* mutation status in tumors from the above *Sftpc^CreER/CreER*; *Trp53^flox/flox*;*Kras^nat/ex3op* mice injected with tamoxifen (termed *Trp53^-/-*), as the heterozygous status of the *Kras^nat/ex3op* background allows for the most direct comparison, to tumors from a parallel control cohort not injected with tamoxifen (termed *Trp53^+/+*) prior to urethane exposure (*Figure 2A* and *Supplementary file 1*). As mentioned above, the *Trp53^flox* alleles were confirmed by PCR to be recombined in tumors from the *Trp53^-/-* background. The same analysis was thus performed on tumors from the *Trp53^+/+* cohort, which identified one tumor having a significant degree of *Trp53^flox* recombination, which was excluded from the analysis of comparing *Trp53^+/+* versus *Trp53^-/-* mice (*Figure 2—figure supplement 1A*). Consistent with the role of p53 as a tumor suppressor during lung tumor progression (*Feldser et al., 2010*; *Junttila et al., 2010*), loss of p53 tracked with larger, although we note not with more tumors (*Figure 2B, Figure 2—figure supplement 1B, C*). To examine the effect of *p53* deficiency on the RAS mutation tropism of urethane, we sequenced *Kras* derived from mRNA isolated from these lung tumors to screen for mutations at the three main hotspots of $G_{12}$, $G_{13}$, and $Q_{61}$. In complete agreement with the previous observation that the increased protein expression of the endogenous *Kras^ex3op* allele shifts the RAS mutation tropism of urethane from the canonical $Q_{61}$ to $G_{12}$ oncogenic mutations (*Pershing et al., 2015*), tumors with an oncogenic mutation in the *Kras^ex3op* allele from control *Trp53^+/+* mice similarly had $G_{12}$ oncogenic mutations in this allele (*Figure 2C*). Having established that urethane behaves identically as previously

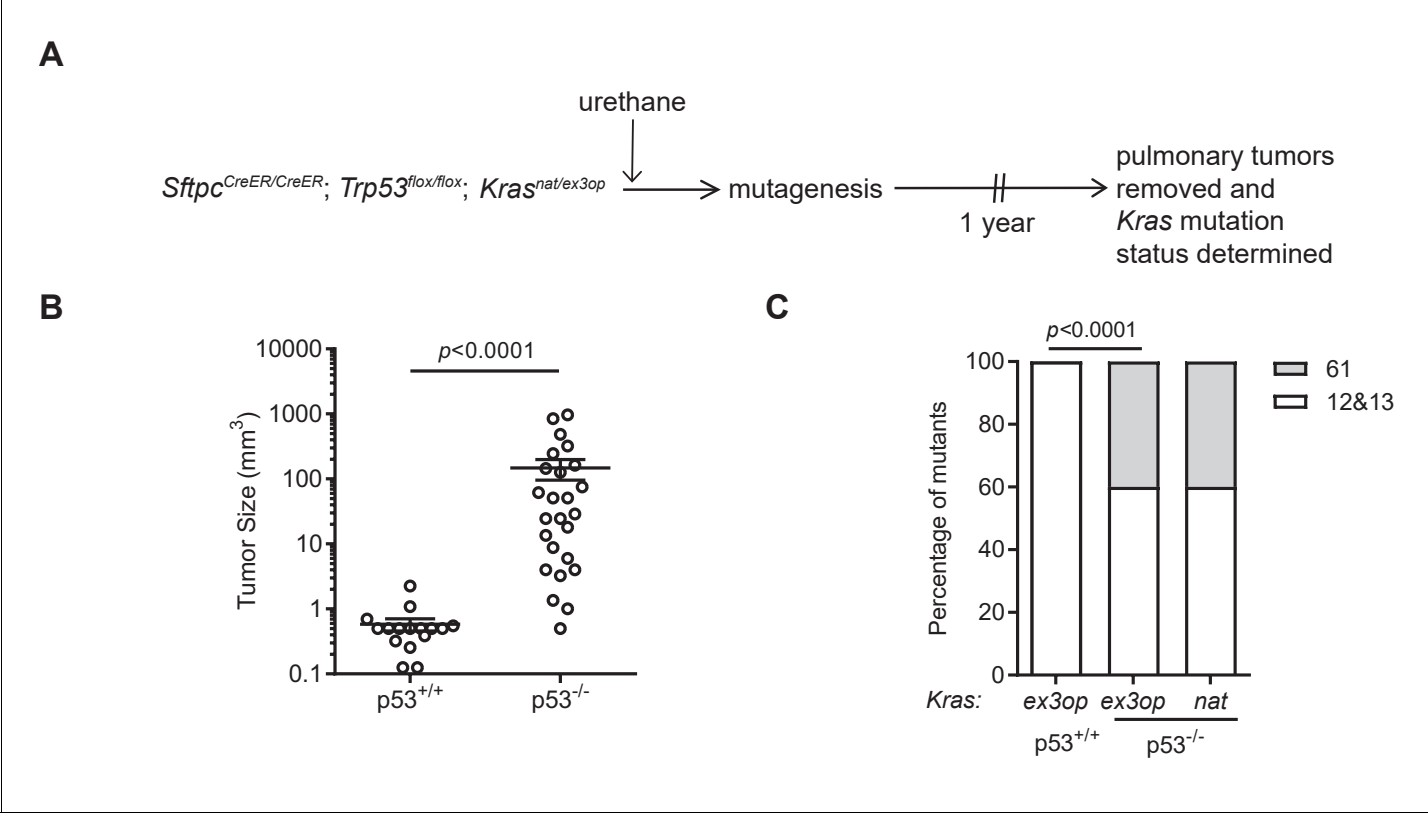

**Figure 2.** Loss of p53 reprograms the RAS mutation tropism of urethane. (A) Experimental design to obtain urethane-induced lung tumors from p53$^{+/+}$ mice. (B) Mean ± SEM of urethane-induced tumor size in $Sftpc^{CreER/CreER}$;$Trp53^{fl/fl}$;$Kras^{ex3op/nat}$ mice not treated (p53$^{+/+}$, n = 16 tumors) or treated with tamoxifen (p53$^{-/-}$, n = 25 tumors). Mann–Whitney test. (C) % of urethane-induced tumors with an oncogenic mutation at codon G$_{12/13}$ (white bar) versus Q$_{61}$ (gray bar) in the $Kras^{nat}$ versus $Kras^{ex3op}$ allele in $Sftpc^{CreER/CreER}$;$Trp53^{fl/fl}$;$Kras^{ex3op/nat}$ mice not treated (p53$^{+/+}$) or treated with tamoxifen (p53$^{-/-}$) where indicated. n = 4 tumors $ex3op$ p53$^{+/+}$, 5 tumors $nat$ p53$^{-/-}$, and 10 tumors $ex3op$ p53$^{-/-}$. Two-sided Fisher's exact test.

The online version of this article includes the following source data and figure supplement(s) for figure 2:

**Source data 1.** Raw data for *Figure 2B, C*, *Figure 2—figure supplement 1A–C*.
**Figure supplement 1.** The effect of p53 loss on tumor burden and multiplicity.

reported in this regard, we turned our attention to the types of mutations recovered in the $Kras^{ex3op}$ allele from the urethane-induced tumors of the $Trp53^{-/-}$ mice. Sequencing revealed that 40% hotspot oncogenic mutations in the $Kras^{ex3op}$ allele were now detected at Q$_{61}$ (*Figure 2C*). These findings are consistent with the loss of p53 partially shifting the oncogenic mutations in the $Kras^{ex3op}$ allele detected in tumors back to the canonical Q$_{61}$ mutations of urethane.

## The mutation signature of urethane is not affected by the $Kras^{ex3op}$ allele

To determine if the shift in oncogenic mutations from Q$_{61}$ in the $Kras^{nat}$ allele to G$_{12}$ in the $Kras^{ex3op}$ allele, and then back again in the $Trp53^{-/-}$ background, resides at the level of the locus or with the amount of encoded protein, we determined whether urethane induces different mutations in these two alleles. To this end, we turned to the ultra-sensitive maximum depth sequencing (MDS) assay (*Jee et al., 2016*), which we adapted for the mammalian genome to detect urethane-induced mutations within days of carcinogen exposure (*Li et al., 2020*). Since only a short region of genomic DNA can be sequenced by this approach, it is not possible to track oncogenic mutations in exon 1 or 2 and also determine the identity of the $Kras$ allele (*native* versus *ex3op*) based on the codon usage in exon 3. Thus, we compared mutations arising in the $Kras^{nat/nat}$ versus $Kras^{ex3op/ex3op}$ genotype. To ensure potent mutagenesis for detection purposes, these two strains were crossed into the pure 129

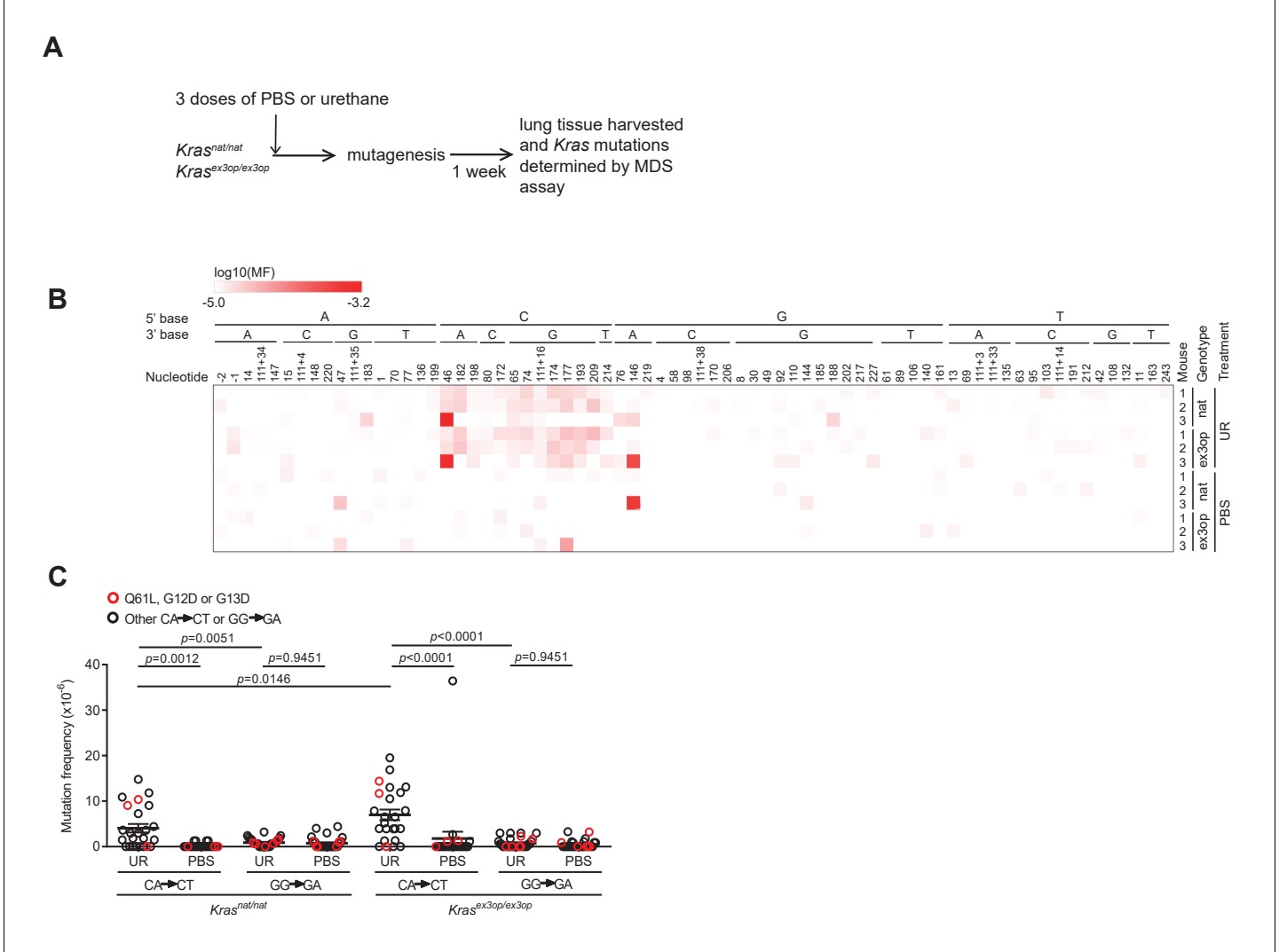

**Figure 3.** The mutation signature of urethane is not affected by the *Kras*^ex3op allele. (A) Experimental design to identify mutations induced by urethane in mouse lung in a *Kras*^nat versus *Kras*^ex3op background. (B) Heatmap of the log-transformed mutation frequency (MF) of A>T transversions determined by maximum depth sequencing (MDS) sequencing the exons 1 and 2 of *Kras* from the lungs of mice exposed to urethane (UR) in a *Kras*^nat/nat (*nat*) (n = 3 mice) versus *Kras*^ex3op/ex3op (*ex3op*) (n = 3 mice) background. Nucleotide number as well as the 5′ and 3′ base of the substituted A are shown at the top; '-' indicates nucleotides upstream of ATG start codon in 5′UTR; '111+' indicates nucleotides in the intron downstream of exon 1. (C) Mean ± SEM mutation frequency of all C*A*>C*T* mutations in *Kras* exon 2, with Q$_{61}$L mutation highlighted in red, as well as all G*G*>G*A* mutations in *Kras* exon 1, with G$_{12}$D and G$_{13}$D mutations highlighted in red, derived from the aforementioned MDS sequencing of *Kras* exons 1 and 2 from the lungs of *Kras*^nat/nat versus *Kras*^ex3op/ex3op mice treated with either urethane or PBS (n = 3 mice each). Holm–Sidak multiple comparisons test following one-way ANOVA. The online version of this article includes the following source data and figure supplement(s) for figure 3:

**Source data 1.** Raw data for *Figure 3C*, *Figure 3—figure supplement 1B, C*.
**Figure supplement 1.** Mutagenesis profile of *Sftpc*^CreER/CreER;*Trp53*^fl/fl;*Kras*^nat/nat and *Sftpc*^CreER/CreER;*Trp53*^fl/fl;*Kras*^ex3op/ex3op mice.

background, which is particularly sensitive to urethane (*Malkinson and Beer, 1983*; *Shimkin and Stoner, 1975*), and mice were injected three times instead of just once as above with either the vehicle PBS or urethane. Seven days later, before overt cell selection (*Li et al., 2020*), the mice were humanely euthanized and their lungs removed and subjected to MDS sequencing to determine both the mutation signature and the type of *Kras* driver mutations induced by urethane (*Figure 3A* and *Supplementary file 2*). Mutation frequencies based on MDS sequencing of *Kras* exons 1 and 2 were averaged for A>T transversions, log$_{10}$ transformed, and displayed in a heatmap format. This

revealed a trend towards A>T transversions within the context of a 5′ C in both the *native* and *ex3op* alleles of *Kras* specifically in urethane-exposed mice (*Figure 3B*), consistent with previously identified bias for urethane (*Li et al., 2020*; *Valentine et al., 2020*; *Westcott et al., 2015*). Moreover, the frequency of these C$\underline{A}$>C$\underline{T}$ mutations, which give rise to $Q_{61}L$ (C$\underline{A}_{182}$A>C$\underline{T}$A), is significantly higher than G$\underline{G}$>G$\underline{A}$ mutations, which give rise to $G_{12}D$ (G$\underline{G}_{35}$T>G$\underline{A}$T) and $G_{13}D$ (G$\underline{G}_{38}$C>G$\underline{A}$C), in both $Kras^{nat/nat}$ and $Kras^{ex3op/ex3op}$ mice (*Figure 3C*). Similar results were found upon repeating the experiment with a single injection of urethane in the less sensitive 129/B6 mixed strain background using $Sftpc^{CreER/CreER}$;$Trp53^{flox/flox}$;$Kras^{nat/nat}$ versus $Sftpc^{CreER/CreER}$;$Trp53^{flox/flox}$; $Kras^{ex3op/ex3op}$ mice treated or not with tamoxifen (*Figure 3—figure supplement 1A*). The only exception was that far fewer C$\underline{A}$>C$\underline{T}$ mutations were detected in general, and perhaps as a consequence, the difference between the frequency of C$\underline{A}$>C$\underline{T}$ and G$\underline{G}$>G$\underline{A}$ mutations was no longer significant, although G$\underline{G}$>G$\underline{A}$ mutations were detected in both the urethane and PBS cohorts, suggesting false positives, although other interpretations are possible (*Figure 3—figure supplement 1B, C*). Thus, with the above proviso, urethane mutagenesis does not appear to be changed at the $Kras^{ex3op}$ locus, at least within the detection limit of the MDS assay. These and the above findings support a model whereby the observed bias towards $G_{12/13}$-driver mutation in the $Kras^{ex3op}$ allele in urethane-induced tumors is a product of negative selection against the biochemically more active $Q_{61}$ oncogenic mutations, rather than a change in the mutational spectrum of urethane.

## p53 loss tracks with an mRNA allelic imbalance of less active oncogenic mutations

Not only was there a shift in the oncogenic mutations in the $Kras^{ex3op}$ allele upon the loss of p53, as noted above, but surprisingly also in the $Kras^{nat}$ allele. In more detail, we found that 60% hotspot mutations in the $Kras^{nat}$ allele of urethane-induced tumors from the $Sftpc^{CreER/CreER}$;$Trp53^{flox/flox}$; $Kras^{nat/ex3op}$ mice in which the $Trp53$ gene was recombined in the lung occurred at codon $G_{12}$, and in one case also $G_{13}$ (*Figure 2C*). We also note that the percentage of tumors that have *Kras* hotspot mutations is higher in p53$^{-/-}$ mice (*Figure 4—figure supplement 1A*). In addition, even though oncogenic mutations are generally more frequent in the $Kras^{ex3op}$ allele, the percentage of tumors with mutations in the $Kras^{nat}$ allele is higher in p53$^{-/-}$ mice (*Figure 4—figure supplement 1B*). One interpretation of these findings is that the absence of p53 enhances the ability of $G_{12/13}$ mutations to be productive. Given the above tight relationship between Kras expression and mutation type, we explored a possible relationship between p53 loss and higher expression of *Kras* alleles with a $G_{12/13}$ mutation. To this end, we calculated the ratio of mutant to wildtype (non-mutant) *Kras* mRNA based on the number of cDNA sequencing reads matched to the mutant or wildtype allele from the above analysis. This revealed a clear demarcation in *Kras* mRNA levels between alleles with a $Q_{61}$ versus a $G_{12/13}$ oncogenic mutation. In detail, a waterfall plot revealed the mutant:wildtype ratio was higher in $G_{12/13}$-mutant *Kras* alleles, with an average ratio of approximately two copies for the mutant *Kras* transcript to every copy of wildtype *Kras* counterpart. Conversely, the mutant:wildtype ratio was lower in $Q_{61}$-mutant *Kras* alleles, with an average ratio of ~0.8 copies for the mutant *Kras* transcript to each copy of wildtype *Kras* counterpart (*Figure 4A, Figure 4—figure supplement 1C, D*). The allelic imbalance in the $G_{12/13}$-mutant tumors appears to be important for tumorigenesis. Namely, cross-referencing the mutant:wildtype *Kras* ratio to the mutation type and size of tumors revealed that $G_{12/13}$-mutant tumors with a mutant:wildtype ratio $\geq 1.5$ are larger, reaching the same size of $Q_{61}$-mutant tumors (*Figure 4B*). These findings support a model whereby *Kras* with less active $G_{12/13}$ oncogenic mutations undergo a selection for higher expression to compensate for the lower signaling, while, if anything, the reverse was seen for the more active $Q_{61}$ oncogenic mutations.

## Allelic mRNA imbalance of $G_{12/13}$-mutant *Kras* alleles tracks with expression of three Ras target genes

To explore the possibility of a selection for a specific degree of signaling, we determined the expression of the genes *Dusp6*, *Egr1*, and *Fosl1* by measuring the levels of encoded mRNA by qRT-PCR. These three genes were chosen as they are all activated by oncogenic RAS through the MAPK pathway (*Buffet et al., 2017*; *Chung et al., 2017*; *Esnault et al., 2017*; *Gillies et al., 2017*; *Kidger and Keyse, 2016*; *McMahon and Monroe, 1995*; *Swarbrick et al., 2008*; *Unni et al., 2018*; *Vallejo et al., 2017a*; *Vallejo et al., 2017b*; *Zhang et al., 2010*), which is the very pathway

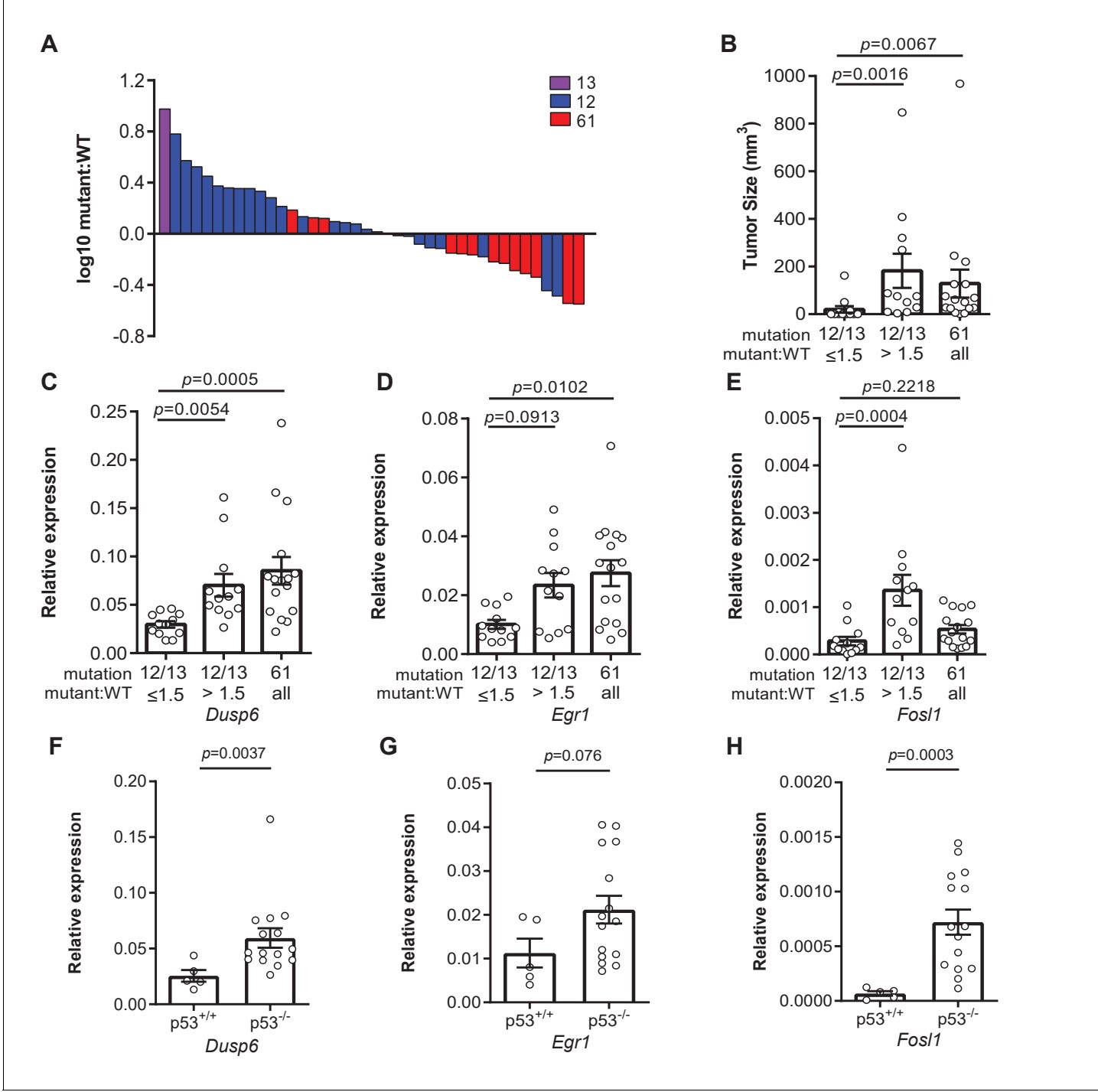

**Figure 4.** Loss of p53 promotes higher expression of weaker oncogenic mutations. (**A**) Log$_{10}$-transformed ratio of mutant to wildtype *Kras* mRNA determined by RT-qPCR in all Kras hotspot-mutant tumors (n = 40) derived from *Figures 1* and *2*. (**B**) Mean ± SEM size of tumors with a G$_{12/13}$ oncogenic Kras mutation with a high (>1.5, n = 12 tumors) versus low (≤1.5, n = 12 tumors) mutant:WT ratio versus tumors with a Q$_{61}$ oncogenic Kras mutation (n = 16 tumors). Dunn's multiple comparison test following Kruskal–Wallis test. (**C–H**) Mean ± SEM levels of the indicated mRNAs normalized to β-actin (relative expression) in (**C–E**) tumors with a G$_{12/13}$ oncogenic Kras mutation with a high (>1.5, n = 12 tumors) versus low (≤1.5, n = 12 tumors) mutant:WT ratio versus tumors with a Q$_{61}$ oncogenic Kras mutation (n = 16 tumors) or (**F–H**) tumors from *Sftpc$^{CreER/CreER}$;Trp53$^{fl/fl}$;Kras$^{ex3op/nat}$* mice not treated (p53$^{+/+}$, n = 5 tumors) or treated with tamoxifen (p53$^{-/-}$, n = 15 tumors) partitioned by p53 mutation status. (**C–E**) Dunn's multiple comparison test following Kruskal–Wallis test. (**F–H**) Mann–Whitney test.

The online version of this article includes the following source data and figure supplement(s) for figure 4:

**Source data 1.** Raw data for *Figure 4A–H*, *Figure 4—figure supplement 1A–K*, *Figure 4—figure supplement 2A, B*.

*Figure 4 continued on next page*

*Figure 4 continued*

**Figure supplement 1.** Allelic imbalance and MAPK signaling in Kras hotspot-mutant tumors.

**Figure supplement 2.** The imbalance at mRNA level could not be fully attributed to the imbalance of DNA copy number.

promoting proliferation (*Drosten and Barbacid, 2020*; *Hymowitz and Malek, 2018*; *Ryan et al., 2015*) or senescence (*Muñoz-Espín and Serrano, 2014*). Binning the relative expression of these three genes into *Kras* $G_{12/13}$-mutant tumors with a high (>1.5) versus low (≤1.5) mutant:wildtype ratio revealed that *Kras* $G_{12/13}$-mutant tumors with a high ratio exhibited higher expression. Furthermore, when we compared the relative expression of these three genes in *Kras* $Q_{61}$-mutant tumors, *Dusp6* and *Egr1* were expressed higher compared to the *Kras* $G_{12/13}$-mutant tumors with a low ratio (*Figure 4C–E*). Plotting the relative expression of these three genes versus the mutant:wildtype cDNA ratio of individual tumors showed that increased expression correlated with the allelic ratio for $G_{12/13}$ mutations (*Figure 4—figure supplement 1E–G*). One interpretation of these results is that an increase in expression of the $G_{12/13}$-mutant *Kras* alleles, as measured by a high mutant:wildtype ratio, manifests as an increase in Ras signaling, as measured by an increase in *Dusp6*, *Egr1*, and *Fosl1* mRNA, matching that achieved by the more potent $Q_{61}$-mutated *Kras* alleles. To assess the effect of p53 on this relationship, we compared *Dusp6*, *Egr1*, and *Fosl1* expression between tumors with and without p53. This revealed higher expression of all three genes in the latter tumors, consistent with the loss of p53 permitting higher Ras signaling (*Figure 4F–H*). p53 loss also tracked with a higher mutant:wildtype allelic ratio in tumors with *Kras* $G_{12/13}$-mutations in the *Kras*^ex3op^ allele (*Figure 4—figure supplement 1H*). Finally, we compared *Dusp6*, *Egr1*, and *Fosl1* mRNA levels in tumors of different sizes, which revealed higher expression in larger tumors, regardless of how this was achieved (*Figure 4—figure supplement 1I–K*). This suggests that the absence of p53 expands the spectrum of driver *Kras* mutations to include less active mutations by permitting or even fostering an increase in their expression. In sum, we suggest that collectively these findings are consistent with the identification of specific oncogenic mutations in different Kras and p53 backgrounds, and favor a model whereby an optimal oncogenic signal selects the type of oncogenic mutation to initiate tumorigenesis in a normal cell.

## Discussion

Here, we show that p53 loss reprograms the RAS mutation tropism of urethane. On one hand, loss of this tumor suppressor shifts the canonical $Q_{61}$ oncogenic mutations that are normally detected in the unperturbed *Kras*^nat^ allele of urethane-induced tumors towards the usually rare $G_{12/13}$ mutations, but in conjunction with an increase in the mutant:wildtype ratio. On the other hand, the loss of p53 shifts the prevalence of $G_{12/13}$ mutations in the codon-optimized and more highly translated *Kras*^ex3op^ allele towards $Q_{61}$ mutations. The one common theme in both these shifts is the potential for higher oncogenic signaling, which argues that the degree of oncogenic signaling dictates the type of mutation conducive to initiate tumorigenesis in normal cells (*Figure 5*). In agreement, the expression of three known RAS downstream target genes was similar between *Kras*^ex3op^ with $Q_{61}$ mutations and *Kras*^nat^ with $G_{12/13}$ mutations when coupled with a high mutant:wildtype ratio, suggesting that expression differences render $Q_{61}$ and $G_{12/13}$ mutations fungible. While there is no question that the level of oncogenic signaling can influence tumorigenesis, what is new here is that the sensitivity of normal cells to oncogenic signaling may underlie the type of oncogenic mutation selected, which is relevant to the RAS mutation tropism of human cancers. We also note that neither the loss of p53 nor the change in codon usage in *Kras* altered the mutation signature of urethane, arguing that a selection for an optimal oncogenic signal supersedes the mutational bias of this carcinogen. Such a finding speaks to the somewhat complexing discordance often observed between the mutation signature of tumors and the corresponding driver mutation in human cancers (*Dietlein et al., 2020*; *Temko et al., 2018*), even in cancers in which the mutational signature can be ascribed to a specific mutagenic process (*Buisson et al., 2019*).

There are three presumptions to the above model (*Figure 5*). First, $Q_{61}$ oncogenic mutations in Kras activate the MAPK pathway more potently than $G_{12/13}$ mutations. In agreement, a $Q_{61}$R

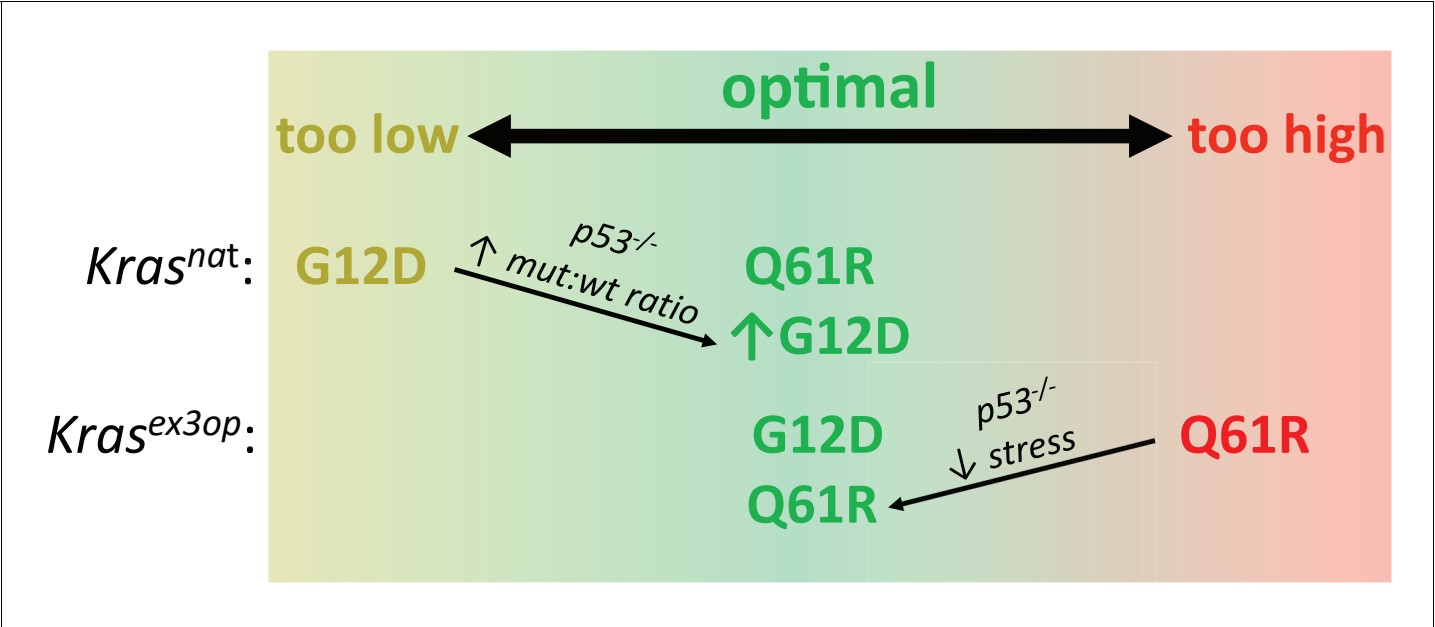

**Figure 5.** Optimal signaling is required for effective tumor initiation. Signaling from a $G_{12}D$ mutation in the native (nat) *Kras* allele and from a $Q_{61}R$ mutation in the codon-optimized (ex3op) *Kras* allele is outside of the window of optimal signaling level achieved by *Kras*$^{nat}$($G_{12}D$) and *Kras*$^{ex3op}$($Q_{61}R$). Loss of p53 alleviates the selection against oncogenic stress and allows the recovery of a $Q_{61}R$ mutation in *Kras*$^{ex3op}$ allele or a $G_{12}D$ mutation in the *Kras*$^{nat}$ allele with elevated mutant:wildtype (mut:wt) mRNA allelic ratio.

mutation in Nras and/or Kras has a much (1000 times) slower intrinsic hydrolysis rate, higher GTP-loading, and more robust activation of the MAPK pathway, and when activated in the skin or hematopoietic system in vivo, is more tumorigenic than the corresponding $G_{12}D$ mutation (**Burd et al., 2014**; **Kong et al., 2016**; **Pershing et al., 2015**). We also find that the expression of three downstream Ras target genes is higher in tumors with a $Q_{61}$ mutation compared to $G_{12/13}$ mutation in the absence of an allelic mRNA imbalance. One caveat, however, is that neither the level of the oncoproteins nor their signaling was directly measured at the protein level as the small amount of tumor material available was devoted to DNA/RNA analysis. Second, increasing Kras level increases signaling. In agreement, the *Kras*$^{ex3op}$ allele has been documented to express more protein in the lung (**Pershing et al., 2015**) and increase ERK phosphorylation in hematopoietic stem cells (**Sasine et al., 2018**). Altering rare codons to common in ectopic or endogenous human *KRAS* also leads to higher translation, KRAS protein expression, and/or MAPK activation (**Ali et al., 2017**; **Fu et al., 2018**; **Lampson et al., 2013**; **Pershing et al., 2015**; **Peterson et al., 2020**). It remains possible, however, that rare codons are differentially translated (**Quax et al., 2015**) or higher Kras protein expression reduces MAPK signaling through negative feedback (**Shin et al., 2009**) specifically in the cell of origin of these pulmonary tumors. We also note the caveats that only three downstream target genes were assayed, and at the transcriptional rather than protein level, and further, that these three genes reside in just one effector pathway of RAS. Third, high oncogenic signaling is a negative selection, and this selection is p53-dependent. It is formally possible that the effect of p53 loss on the mutational bias of urethane is linked to cell fate or other factors instead of suppressing oncogene stress, but this would have to occur very quickly since urethane exposure shortly followed tamoxifen injection. We also note that oncogenic stress and p53 induction could not be measured in individual normal cells immediately after acquiring different oncogenic mutations by urethane mutagenesis, but both these points are well supported in multiple model systems (**Muñoz-Espín and Serrano, 2014**). As such, it is reasonable to propose that the level of oncogenic signaling is a driving force in the selection of the type of oncogenic Kras mutations initiating tumorigenesis (**Figure 5**). Capturing signaling in individual normal cells by single-cell transcriptomics

immediately after induction of different oncogenic Kras mutations may provide a platform to interrogate this model. Further, application of CRISPR-Cas9-based barcoding tools that allow simultaneous lineage tracing and transcriptome analysis of individual cells within a tumor (**Bowling et al., 2020**; **Quinn et al., 2021**) could provide a clearer picture of the signaling dynamics throughout the course of tumor evolution.

We note that urethane has a bias towards C$\underline{A}$N'C$\underline{T/G}$N mutations that are absent in both strand orientations at codons encoding $G_{12}$ and $G_{13}$, and the mutagenic signature of urethane was the same in the $Kras^{nat}$ and $Kras^{ex3op}$ alleles. This suggests that $G_{12/13}$ oncogenic mutations are extremely rare, with urethane either inducing these mutations directly at a low frequency or inducing a cooperating mutation or non-mutational event that promotes a rare, pre-existing Kras-mutant cell to expand. In either scenario, the very fact that tumors arise with $G_{12/13}$ oncogenic mutations argues for potent selection of an extremely rare oncogenic mutation over the far more common mutations favored by the carcinogen. Furthermore, if these mutations are indeed pre-existing, such a finding elevates the importance of promotional events (secondary mutations, inflammation, etc.) in the process of tumor initiation (**Balmain, 2020**). Perhaps related, it is interesting that the loss of p53 did not shift all the oncogenic mutations in $Kras^{ex3op}$ from $G_{12}$ to $Q_{61}$, which is unexpected given the rarity of $G_{12}$ mutations induced by urethane. One interpretation is that oncogenic $G_{12}$ mutations may be superior to $Q_{61}$ mutations in some manner, which is interesting insofar as $G_{12}$ mutations are more common in KRAS than $Q_{61}$ mutations in human cancers (**Prior et al., 2020**).

We also note an imbalance in the mRNA ratio of the mutant:wildtype $Kras$ alleles in some urethane-induced tumors. Amplification of oncogenic $Kras$ alleles has been previously documented in various mouse cancer backgrounds (**Chung et al., 2017**; **Mueller et al., 2018**; **Westcott et al., 2015**). What is new, however, is that this tracked with the type of oncogenic mutation, and even more interesting, in opposite directions. Namely, we observed a bias towards a high mutant:wildtype mRNA ratio in $G_{12/13}$-mutant $Kras$ alleles in the absence of p53. Few tumors show amplification of the $G_{12/13}$-mutant allele based on genomic copy number analysis (**Figure 4—figure supplement 2**), pointing towards transcriptional or post-transcriptional mechanisms. One interpretation of these findings is that loss of p53 allows expansion of tumors with $G_{12/13}$-mutant $Kras$ alleles that are more highly expressed, an observation borne out by signaling analysis. Increasing MAPK signaling by paradoxical activation with a BRAF inhibitor expands the number of cell types that are tumorigenic upon activation of the $LSL$-$Kras^{G12D}$ allele in the lung (**Cicchini et al., 2017**), hence it is quite possible that a similar phenomenon is at play here as well. Such higher expression could be a product of natural variation of gene expression, a mutation caused by urethane, or p53 loss itself.

We also observed a bias towards a low mutant:wildtype mRNA ratio in $Q_{61}$-mutant $Kras$ alleles. Such a finding is consistent with this mutation inducing high oncogenic signaling, and hence a selection against increased expression. Interestingly, this ratio was low irrespective of the p53 status. Perhaps there is an upper limit to oncogenic signaling, even in the absence of p53, that is tumor-promoting. In support, we note that $G_{12/13}$-mutant tumors with a high mutant:wildtype mRNA ratio had the same Ras signaling output as $Q_{61}$-mutant tumors, at least as assessed by the expression level of the three tested genes downstream of Ras. Indeed, RAS-mutant human tumors, which often have disruption of p53 and other tumor suppressors, typically do not have BRAF or EGFR mutations and vice versa outside of drug resistance settings (**Cancer Genome Atlas Research Network et al., 2018**). Moreover, co-expressing oncogenic RAS with oncogenic BRAF or EGFR can be growth suppressive (**Cisowski et al., 2016**; **Unni et al., 2018**).

In conclusion, multiple independent lines of investigation support a model whereby the sensitivity of a normal cell to a narrow range of oncogenic signaling dictates the nature of the mutation conducive to tumor initiation. Too low, a mutation fails to induce proliferation, as supported by the finding that less active $G_{12}$ mutants are rarely recovered in the $Kras^{nat}$ allele unless accompanied by an allelic imbalance permitted or fostered by the absence of p53. Too high, a mutation induces arrest, as supported by the finding that more active $Q_{61}$ mutants are not recovered from the more highly expressed $Kras^{ex3op}$ allele unless oncogene-induced senescence is suppressed by the loss of p53 (**Figure 5**). Moreover, this selection pressure exceeds the mutational bias of the mutagen, as supported by the observations that $G_{12}$ mutations fail to match the urethane mutational signature, yet nevertheless arise at a high frequency in urethane-induced tumors in a $Kras^{ex3op}$ background. Such findings speak to the extreme mutational bias of

urethane-induced tumors, and perhaps more broadly, provide mechanistic insight into the RAS mutation tropism observed in human cancers.

# Materials and methods

## Key resources table

| Reagent type (species) or resource | Designation | Source or reference | Identifiers | Additional information |
|---|---|---|---|---|
| Strain, strain background (*Mus musculus*) | 129 background - $Kras^{ex3op}$ | *Pershing et al., 2015*, MMRRC | Stock 050601-UNC; MGI:5708830 | |
| Strain, strain background (*Mus musculus*) | B6.129 mixed background - $Trp53^{flox/flox}$ | The Jackson Laboratory | Stock 008462; MGI:1931011 | |
| Strain, strain background (*Mus musculus*) | B6.129 mixed background - $Sftpc^{CreER/CreER}$ | *Xu et al., 2012*, gift from Mark Onaitis | MGI:5305340 | |
| Strain, strain background (*Mus musculus*) | B6.129 mixed background - $Sftpc^{CreER/CreER}$, $Trp53^{flox/flox}$;$Kras^{ex3op}$ | This paper | N/A | See Materials and methods section 'Mice' |
| Sequence-based reagent | $Kras^{ex3op}$ genotyping F | This paper | PCR primers | TGGTAGGGTAGAAACTAGGATTC |
| Sequence-based reagent | $Kras^{ex3op}$ genotyping R | This paper | PCR primers | GAGTACACAGAGAGACCATTTCAAC |
| Sequence-based reagent | $Trp53$ genotyping F | This paper | PCR primers | CACAAAAAACAGGTTAAACCCA |
| Sequence-based reagent | $Trp53$ genotyping WT R | This paper | PCR primers | AGCACATAGGAGGCAGAGAC |
| Sequence-based reagent | $Trp53$ genotyping Del R | This paper | PCR primers | GAAGACAGAAAAGGGGAGGG |
| Sequence-based reagent | $Sftpc$ genotyping F | This paper | PCR primers | GCTTCACAGGGTCGGTAG |
| Sequence-based reagent | $Sftpc$ genotyping R | This paper | PCR primers | GAGGCACCGCTCCGCGAG |
| Sequence-based reagent | $Sftpc$ genotyping CreER R | This paper | PCR primers | CAACTCACAACGTGGCACTG |
| Sequence-based reagent | Tumor sequencing primers | This paper | PCR primers | *Supplementary file 3* |
| Sequence-based reagent | qPCR primers | This paper | PCR primers | *Supplementary file 3* |
| Sequence-based reagent | MDS assay primers | This paper | PCR primers | *Supplementary file 3* |
| Peptide, recombinant protein | Proteinase K | New England Biolabs | Cat# P8107S | |
| Peptide, recombinant protein | RNase A | Sigma | Cat# R4642 | |
| Peptide, recombinant protein | EcoRV | New England Biolabs | Cat# R3195 | |
| Peptide, recombinant protein | EcoRI | New England Biolabs | Cat# R3101 | |
| Peptide, recombinant protein | XmnI | New England Biolabs | Cat# R0194 | |
| Peptide, recombinant protein | Exonuclease I | New England Biolabs | Cat# M0293 | |
| Commercial assay or kit | iScript cDNA synthesis kit | Bio-Rad | Cat# 1708890 | |

*Continued on next page*

*Continued*

| Reagent type (species) or resource | Designation | Source or reference | Identifiers | Additional information |
|---|---|---|---|---|
| Commercial assay or kit | Platinum Taq Polymerase | Thermo Fisher Scientific | Cat# 10966083 | |
| Commercial assay or kit | QIAquick PCR Purification Kit | Qiagen | Cat# 28104 | |
| Commercial assay or kit | Q5 Hot Start High-Fidelity DNA Polymerase | New England Biolabs | Cat# M0493 | |
| Commercial assay or kit | Agencourt AMPure XP | Beckman Coulter | Cat# A63880 | |
| Commercial assay or kit | iTaq Universal SYBR Green Supermix | Bio-Rad | Cat# 1725120 | |
| Commercial assay or kit | PrimeTime Gene Expression Master Mix | Integrated DNA Technologies | Cat# 1055770 | |
| Chemical compound, drug | Tamoxifen | Sigma | Cat# T5648 | |
| Chemical compound, drug | Corn oil | Sigma | Cat# C8267 | |
| Chemical compound, drug | Urethane | Sigma | Cat# U2500 | |
| Chemical compound, drug | RLT buffer | Qiagen | Cat# 79216 | |
| Chemical compound, drug | β-Mercaptoethanol | Thermo Fisher Scientific | Cat# 21985023 | |
| Chemical compound, drug | Trizol LS | Thermo Fisher Scientific | Cat# 10296010 | |
| Chemical compound, drug | dNTP | New England Biolabs | Cat# N0447S | |
| Chemical compound, drug | Agarose | EMD Millipore | Cat# 2120-OP | |
| Chemical compound, drug | Tris | EMD Millipore | Cat# 9210-OP | |
| Chemical compound, drug | EDTA | VWR | Cat# 97061–406 | |
| Chemical compound, drug | Sodium dodecyl sulfate | Sigma | Cat# L4509 | |
| Chemical compound, drug | Phenol | Sigma | Cat# P1037 | |
| Chemical compound, drug | Chloroform | Macron Fine Chemicals | Cat# 4440-04 | |
| Chemical compound, drug | Ethanol | VWR | Cat# 89125-190 | |
| Chemical compound, drug | 10X exonuclease I buffer | New England Biolabs | Cat# B0293S | |
| Chemical compound, drug | Sodium chloride | EMD Millipore | Cat# SX0420 | |
| Chemical compound, drug | Potassium chloride | VWR | Cat# BDH0258 | |
| Chemical compound, drug | Potassium phosphate monobasic | Sigma | Cat# 795488 | |
| Chemical compound, drug | Sodium phosphate dibasic | Sigma | Cat# RDD022 | |
| Software, algorithm | fastq-join | https://usegalaxy.org/ | Galaxy Version 1.1.2–806.1 | See Materials and methods section 'Sequencing data analysis' |

*Continued on next page*

*Continued*

| Reagent type (species) or resource | Designation | Source or reference | Identifiers | Additional information |
|---|---|---|---|---|
| Software, algorithm | Filter by Quality | https://usegalaxy.org/ | Galaxy Version 1.0.0 | See Materials and methods section 'Sequencing data analysis' |
| Software, algorithm | Trim | https://usegalaxy.org/ | Galaxy Version 0.0.1 | See Materials and methods section 'Sequencing data analysis' |
| Software, algorithm | Filter sequences by length | https://usegalaxy.org/ | Galaxy Version 1.1 | See Materials and methods section 'Analysis of MDS data' |
| Software, algorithm | Group | https://usegalaxy.org/ | Galaxy Version 2.1.4 | See Materials and methods section 'Sequencing data analysis' |
| Software, algorithm | Barcode Splitter | https://usegalaxy.org/ | Galaxy Version 1.0.0 | See Materials and methods sections 'Sequencing data analysis' and 'Analysis of MDS data' |
| Software, algorithm | PEAR pair-end read merger | *Zhang et al., 2014b* | Version 0.9.8 | https://cme.h-its.org/exelixis/web/software/pear/ |
| Software, algorithm | Morpheus | https://software.broadinstitute.org/morpheus | N/A | Generation of heatmaps |
| Software, algorithm | Prism | GraphPad | Version 6 | |

## Mice

$Kras^{ex3op}$ mice were previously described (*Pershing et al., 2015*), $Trp53^{flox/flox}$ mice (*Marino et al., 2000*) were purchased from The Jackson Laboratory (Jax # 008462), and $Sftpc^{CreER/CreER}$ mice (*Xu et al., 2012*) were kindly provided by Mark Onaitis (University of California at San Diego). $Sftpc^{CreER/CreER};Trp53^{flox/flox}; Kras^{ex3op}$ mice were from a mixed 129 × C57BL/6 background. $Kras^{ex3op}$ mice used for mutagenesis studies were from a pure 129 background. Mice were genotyped using the following primers:

*Kras native* and *ex3op* alleles:

> *Kras* F: 5′-TGGTAGGGTAGAAACTAGGATTC-3′
> *Kras* R: 5′-GAGTACACAGAGAGACCATTTCAAC-3′
> Products: 504 bp ($Kras^{ex3op}$) or 614 bp ($Kras^{nat}$)

$Sftpc^{CreER}$ allele:

> *Sftpc* F: 5′-GCTTCACAGGGTCGGTAG-3′
> *Sftpc* R: 5′-GAGGCACCGCTCCGCGAG-3′
> *Spc-CreER* R: 5′-CAACTCACAACGTGGCACTG-3′
> Products: 550 bp (*Sftpc*) or 500 bp ($Sftpc^{CreER}$)

*Trp53* allele:

> *Trp53* F: 5′-CACAAAAAACAGGTTAAACCCA-3′
> *Trp53* WT R: 5′-AGCACATAGGAGGCAGAGAC-3′
> *Trp53* Del R: 5′-GAAGACAGAAAAGGGGAGGG-3′
> Products: 288 bp (wildtype *Trp53*), 370 bp (unrecombined $Trp53^{fl}$), and 612 bp (recombined $Trp53^{fl}$).

All animal experiments were approved by Duke IACUC.

## Tumor experiments

A 20 mg/ml tamoxifen (Sigma T5648) solution was made by dissolving tamoxifen in corn oil. Six- to eight-week-old mice of the indicated genotypes were injected intraperitoneally with 0.25 mg/g body weight tamoxifen every other day for four doses. Urethane (Sigma U2500) was dissolved in PBS and injected intraperitoneally at dose of 1 mg/g body weight 1 week after the last injection of tamoxifen. Approximately 12 months after urethane injection, all mice were humanely euthanized, after which lung tumors were counted, measured, and microdissected for RNA and DNA extraction. The 12-month time point was chosen based on similar previous studies (*Dwyer-Nield et al., 2010*;

*Gurley et al., 2015*; *Miller et al., 2003*; *You et al., 1989*) but also to take into account a potentially longer tumor latency in the mixed 129 × C57BL/6 background (*Malkinson and Beer, 1983*; *Shimkin and Stoner, 1975*). Tumor volume was calculated as ½ (length × width$^2$). Tumor burden was calculated as the sum of tumor volumes.

## Mutagenesis experiments

Six- to eight-week-old *Kras$^{nat/nat}$* or *Kras$^{ex3op/ex3op}$* mice were intraperitoneally injected daily for 3 days with either urethane or PBS as above. These mice were humanely euthanized 1 week later, and the lungs collected for the extraction of genomic DNA. Alternatively, 6- to 8-week-old *Sftpc$^{CreER/CreER}$;Trp53$^{flox/flox}$;Kras$^{nat/nat}$* or *Sftpc$^{CreER/CreER}$;Trp53$^{flox/flox}$;Kras$^{ex3op/ex3op}$* mice were or were not injected intraperitoneally with 0.25 mg/g body weight tamoxifen every other day for four doses. One week after the last injection of tamoxifen, the same mice were intraperitoneally injected with one dose of either urethane or PBS. The mice were humanely euthanized 1 month later, and the lungs collected for the extraction of genomic DNA.

## RNA and DNA extraction

Tumors or normal lung tissues were lysed in RLT buffer (Qiagen 79216) with 1% β-mercaptoethanol (Thermo Fisher Scientific 21985023) and 5 units/ml proteinase K (New England Biolabs P8107S) at 55℃ for 30 min. RNA and DNA were then extracted from the lysate using Trizol LS (Thermo Fisher Scientific 10296010) following the manufacturer's instructions. RNA was subsequently converted to cDNA using iScript cDNA synthesis kit (Bio-Rad 1708890) following the manufacturer's instructions.

## PCR and sequence analysis of tumors

*Kras* exons 1–3 were amplified from the cDNA of tumor or normal lung tissue using nested PCR. Primers are listed in *Supplementary file 3*. PCR reactions were as follows:

### PCR1

1 µl cDNA, 0.5 µl of 10 µM forward primer and 0.5 µl reverse primer, 2 µl of 2.5 mM dNTP (New England Biolabs N0447S), 0.75 µl of 50 mM MgCl$_2$, 2 µl of 5X buffer, and 0.1 µl Platinum Taq Polymerase (Thermo Fisher Scientific 10966083) in a total volume of 25 µl. PCR cycles were as follows: one cycle at 94℃ for 2 min and 25 cycles at 94℃ for 30 s, 56℃ for 30 s, and 72℃ for 30 s.

### PCR2

2.5 µl of PCR1 reaction, 0.5 µl of 10 µM forward primer and 0.5 µl reverse primer, 2 µl of 2.5 mM dNTP, 0.75 µl of 50 mM MgCl$_2$, 2 µl of 5X buffer, and 0.1 µl Platinum Taq Polymerase in a total volume of 25 µl. PCR cycles were as follows: one cycle at 94℃ for 2 min and 20 cycles at 94℃ for 30 s, 58℃ for 30 s, and 72℃ for 35 s.

10 µl product for PCR2 was then analyzed with gel electrophoresis to check PCR efficiency. For samples with no product, *Kras* exons 1 and 2 were amplified from the DNA of the same tumor separately using nested PCR. Primers are listed in *Supplementary file 3*. PCR reactions were as follows:

### PCR1

1 µl cDNA, 0.5 µl of 10 µM forward primer and 0.5 µl reverse primer, 2 µl of 2.5 mM dNTP, 0.75 µl of 50 mM MgCl$_2$, 2 µl of 5X buffer, and 0.1 µl Platinum Taq Polymerase in a total volume of 25 µl. PCR cycles were as follows: one cycle at 94℃ for 2 min and 25 cycles at 94℃ for 30 s, 53℃ for 30 s, and 72℃ for 15 s.

### PCR2

The reactions comprised 5 µl PCR1, 0.5 µl of 10 µM forward primer and 0.5 µl reverse primer, 2 µl of 2.5 mM dNTP, 0.75 µl of 50 mM MgCl$_2$, 2 µl of 5X buffer, and 0.1 µl Platinum Taq Polymerase in a total volume of 25 µl. PCR cycles were as follows: one cycle at 94℃ for 2 min and 20 cycles at 94℃ for 30 s, 57℃ for 30 s, and 72℃ for 18 s.

Products from PCR2 were pooled and purified with Ampure XP beads according to the manufacturer's protocol (Beckman Coulter A63880). The library was sequenced using MiSeq v2 Nano 250 PE

at Duke Center for Genomic and Computational Biology. All primers were synthesized by Integrated DNA Technologies.

## Sequencing data analysis

Raw sequencing data were uploaded to usegalaxy.org (*Jalili et al., 2020*). For analysis of amplicon from cDNA and exon 2 from genomic DNA, paired-end reads were joined with fastq-join tool, with the maximum percentage difference between matching segments set at 20% and the minimum length of matching segments set at 10. For analysis of amplicon from exon 1 from genomic DNA, only read 1 was used. The joined-read for cDNA and exon 2, or read 1 for exon 1, was then processed with Filter by Quality Tool, requiring 75% of bases have quality equal to or higher than the cut-off value of 20. For the analysis of amplicon from cDNA, the forward and reverse index was then extracted from the filtered reads through the Trim Tool. Bases encoding codons 12, 13, and 61, as well as nucleotide 96 (where a SNP exists that differentiates 129 from B6 strains) and 8 nucleotides in the middle of exon 3 (where the sequence differs between native and ex3op versions of *Kras*), where appropriate were also extracted through the Trim Tool. Collapsing by the index (representing individual samples) and counting the reads for each variant of the extracted region of interest were then performed by the Group Tool. Mutation and allele information were then assigned to each extracted region of interest by comparing the sequences of the extracted region of interest with reference sequences for all possible missense mutations in codons 12, 13, and 61, SNP at nucleotide 96, and either native or codon-optimized exon 3 using the Excel program. The fraction of each variant of the extracted region was calculated by dividing the counts for that variant by the total number of counts per sample. For analysis of amplicon from genomic DNA, quality filtered-reads were split into separate files using the Barcode Splitter Tool and 5' index. Each of the files were then trimmed from 3' end to expose the 3' index. Trimmed files were further split into separate files corresponding to different samples using the Barcode Splitter Tool and 3' index. For each sample, bases covering codons 12 and 13 in exon 1 and codon 61 in exon 2 were extracted through the Trim Tool. The mutation was assigned to extracted bases by comparing them against reference sequences for all possible missense mutations in codons 12, 13, and 61 using the Excel program. The fraction of each variant was calculated by dividing the counts for that variant by the total number of counts per sample. Samples with less than 30 total reads or variants with fraction less than 8% were excluded from analysis.

## qPCR analysis of *Trp53* recombination

Quantitative PCR (qPCR) reactions were performed using iTaq Universal SYBR Green Supermix (Bio-Rad 1725120) and CFX384 touch real-time PCR detection system (Bio-Rad). Reactions comprised 200 ng gDNA, 5 µl Supermix, 0.5 µl forward primer, 0.5 µl reverse primer in a total volume of 10 µl. PCR conditions were one cycle at 95℃ for 3 min, 40 cycles at 95℃ for 10 s, 55℃ for 30 s, and a melt curve cycle (65–95℃ at 0.5℃ increments at 5 s/step). Primers were designed to detect unrecombined *Trp53*<sup>flox</sup> allele (p53 WT), recombined *Trp53*<sup>flox</sup> allele (p53 Del), and the reference gene *Tflc*. Primer sequences were:

> *Trp53* WT and Del F: 5'-ATCCTTTATTCTGTTCGATAAGCTTG-3'
> *Trp53* WT R: 5'-AGGACTACACAGAGAAACCCT-3'
> *Trp53* Del R: 5'-GCTATTGTAGCTAGAACTAGTGGAT-3'
> *Tflc* F: 5'-ACCAAATGGTTCGTACAGCA-3'
> *Tflc* R: 5'-ATGACAGTAGTTTGCTGTTATACATC-3'

The relative levels of *Trp53* WT and *Trp53* Del genomic DNA were calculated using ΔCt method in comparison to *Tflc*. The fraction of *Trp53* Del was then calculated by dividing the relative level of *Trp53* Del by the sum of the relative level of *Trp53* WT and *Trp53* Del. qPCR analysis of *Kras* copy number qPCR reactions was performed using PrimeTime Gene Expression Master Mix (Integrated DNA Technologies 1055770) and CFX384 touch real-time PCR detection system (Bio-Rad). The reactions comprised 200 ng gDNA, 5 µl master mix, and 0.5 µl forward primer, 0.5 µl reverse primer, 0.25 µl probe for *Kras*<sup>nat</sup> allele, *Kras*<sup>ex3op</sup> allele, and reference gene *Tert* in a total volume of 10 µl. The conditions were one cycle at 95℃ for 3 min, 40 cycles at 95℃ for 10 s, 57℃ for 30 s, and a melt curve (65–95℃ at 0.5℃ increments at 5 s/step).

Primers sequences were:

*Kras^nat* F: 5'-GGAATAAGTGTGATTTGCCTTCT-3'
*Kras^nat* R: 5'-5'-ACCTGTCTTGTCTTTGCTGA-3'
*Kras^ex3op* F: 5'-AAGTGCGACCTCCCTAGC-3'
*Kras^ex3op* R: 5'-CTGTCTTGTCTTGGCGCT-3'
*Tert* F: 5'-CCTGACCATCTGGTGACAC-3'
*Tert* R: 5'-GTGCCTTCTCAGAGAACACA-3'

Probe sequences were:

*Kras^nat*: 5'-/5Cy5/AACAGTAGA/TAO/CACGAAACAGGCTCAGGA/3IAbRQSp /- 3'
*Kras^ex3op*: 5'-/5HEX/AACCGTGGA/ZEN/CACCAAGCAGGCC/3IABkFQ /- 3'
*Tert*: 5'-/56-FAM/TGGAACCAA/ZEN/ACATACATGCAGGTGCAG/3IABkFQ /- 3'

The relative levels of *Kras^nat* and *Kras^ex3op* alleles were calculated using ΔCt method in comparison to *Tert*. The copy number of *Kras^nat* and *Kras^ex3op* allele was then determined by comparing the relative DNA level in tumor samples to normal lung tissues from *Kras^nat/nat* or *Kras^ex3o/ex3op* mice, which were used as reference for two copies of *Kras^nat* or *Kras^ex3op* allele, respectively.

qPCR analysis of MAPK signaling qPCR reactions was performed using iTaq Universal SYBR Green Supermix (Bio-Rad 1725120) and CFX384 touch real-time PCR detection system (Bio-Rad). The reactions comprised 1 μl gDNA, 5 μl Supermix, 0.5 μl forward primer, 0.5 μl reverse primer in a total volume of 10 μl. The conditions were one cycle at 95°C for 3 min, 40 cycles at 95°C for 10 s, 58°C for 30 s, and a melt curve (65–95°C at 0.5°C increments at 5 s/step). The primers sequences were:

*Dusp6* F: 5'-ACTTGGACGTGTTGGAAGAGT-3'
*Dusp6* R: 5'-GCCTCGGGCTTCATCTATGAA-3'
*Egr1* F: 5'-CCTGACCACAGAGTCCTTTTCT-3'
*Egr1* R: 5'-AGGCCACTGACTAGGCTGA-3'
*Fosl1* F: 5'-CAGGAGTCATACGAGCCCTAG-3'
*Fosl1* R: 5'-GCCTGCAGGAAGTCTGTCAG-3'
*Actin* F: 5'-CGTGAAAAGATGACCCAGATCATGT-3'
*Actin* R: 5'-CGTGAGGGAGAGCATAGCC-3'

Gene expression values were calculated using the comparative Ct (-ΔΔCt) method (*Livak and Schmittgen, 2001*), using actin as internal control.

## Isolation of genomic DNA for maximum depth sequencing assay

As adapted from *Li et al., 2020*. Lung tissues were cut into fine pieces and resuspended in 500 μl lysis buffer 100 mM NaCl, 10 mM Tris pH 7.6, 25 mM EDTA pH 8.0, and 0.5% SDS in $H_2O$, supplemented with 20 μg/ml RNase A (Sigma R4642). Samples were incubated at 37°C for 1 hr. 2.5 μl of 800 U/ml proteinase K (New England Biolabs P8107S) was then added to each sample, then the samples were vortexed and incubated at 55°C overnight. Genomic DNA was isolated by phenol/chloroform extraction followed by ethanol precipitation using standard procedures.

## Maximum depth sequencing

This method was adapted from published protocols (*Jee et al., 2016*; *Li et al., 2020*). In detail, 20–50 μg of genomic DNA was incubated with EcoRV (New England Biolabs R3195), EcoRI (New England Biolabs R3101), and XmnI (New England Biolabs R0194) for the analysis of the non-transcribed strand of *Kras* exons 1 and 2. Reaction conditions were 5 units of each of the indicated restriction enzymes and per 1 μg DNA per 20 μl reaction. Digested genomic DNA was column purified using QIAquick PCR Purification Kit following the manufacturer's protocol (Qiagen 28104) and resuspended in ddH₂O (35 μl H₂O per 10 μg DNA). The barcode and adaptor were added to the target DNA by incubating purified DNA with the appropriate barcode primers (see below) for one cycle of PCR. PCR reactions comprised 10 μg DNA, 2.5 μl of 10 μM barcode primer (see below), 4 μl of 2.5 mM dNTP, 10 μl of 5X buffer, and 0.5 μl Q5 Hot Start High-Fidelity DNA Polymerase (New England Biolabs M0493) in a total volume of 50 μl. The number of PCR reactions was scaled according to the amount of DNA. PCR conditions were 98°C for 1 min, 60°C for 15 s, and 72°C for 1 min using the barcoding primer for exon 2, followed by the addition of the barcoding primer for exon 1 to the same reaction, 98°C for 1 min, 68°C for 15 s, and 72°C for 1 min. 1 μl of 20,000 U/ml exonuclease I (New

England Biolabs M0293) and 5 µl of 10X exonuclease I buffer (New England Biolabs B0293S) was then added to each 50 µl reaction to remove unused barcoded primers and incubated at 37°C for 1 hr and then 80°C for 20 min. Processed DNA were column-purified using QIAquick PCR Purification Kit as above and resuspended in ddH$_2$O (35 µl H$_2$O per column). The concentration of purified product was measured with SimpliNano spectrophotometer (GE Healthcare Life Sciences). Samples were linear amplified with forward adaptor primer (see below). PCR reactions comprised 1.5 µg DNA, 2.5 µl of 10 µM forward-adaptor primer, 4 µl of 2.5 mM dNTP, 10 µl of 5X buffer, and 0.5 µl Q5 Hot Start High-Fidelity DNA Polymerase in a total volume of 50 µl. The number of PCR reactions was scaled according to the amount of DNA. PCR conditions were as follows: 12 cycles of 98°C for 15 s, 70°C for 15 s, and 72°C for 10 s. 2.5 µl of 10 µM exon-specific reverse primers (see below) and 2.5 µl of 10 µM reverse-adaptor primer (see below) were then added to each 50 µl reaction. The mixtures were then subjected to exponential amplification. PCR conditions were as follows: 4 cycles of 98°C for 15 s, 62°C for 15 s, 72°C for 10 s, 20 cycles of 98°C for 15 s, 70°C for 15 s, and 72°C for 10 s. The final library size was selected and purified with Ampure XP beads according to the manufacturer's protocol (Beckman Coulter A63880). Sequencing was performed using NovaSeq 6000 S Prime 150 bp PE at Duke Center for Genomic and Computational Biology.

## Primers for MDS

As adapted from *Li et al., 2020*.

> Barcode primer: [Forward adaptor][Index][Barcode][Primer]
> Where
> [Forward adaptor] =
> 5′-TACGGCGACCACCGAGATCTACACTCTTTCCCTACACGACGCTCTTCCGATCT-3′
> [Index] = variable length of known sequences from 0 to 7 nucleotides (*Supplementary file 3*)
> [Barcode] = NNNNNNNNNNNNNNN
> *Kras* exon 1 [Primer]=5′-ATCTTTTTCAAAGCGGCTGGCT-3′
> *Kras* exon 2 [Primer]=5′-TCTTCAAATGATTTAGTATTATTTATGGC-3′
> Forward-adaptor primer: 5′-AATGATACGGCGACCACCGAGAT-3′
> Exon-specific reverse primer: [Reverse adaptor][Index][Primer]
> Where
> [Reverse adaptor] =
> 5′-CAAGCAGAAGACGGCATACGAGATGTGACTGGAGTTCAGACGTGTGCTCTTCCGATCT-3′
> [Index] = variable length of known sequences from 0 to 7 nucleotides (*Supplementary file 3*)
> *Kras* exon 1 [Primer]=5′-TATTATTTTTATTGTAAGGCCTGCTGA-3′
> *Kras* exon 2 [Primer]=5′-GACTCCTACAGGAAACAAGT-3′
> Reverse-adaptor primer: 5′-CAAGCAGAAGACGGCATACGAGA-3′

All primers were synthesized by Integrated DNA Technologies.

## Analysis of MDS data

As adapted from *Li et al., 2020*. Raw data were uploaded Galaxy Cloudman (*Jalili et al., 2020*). Read 1 and read 2 were joined via PEAR pair-end read merger (*Zhang et al., 2014b*). The reads were then filtered by quality by requiring 90% of bases in the sequence to have a quality core $\geq$ 20. Filtered reads were split into different files based on assigned sample indexes and variation in sequence lengths using the Barcode Splitter tool and Filter sequences by length tool. The reads were trimmed down to the barcode and the target exon. Trimmed reads were grouped by barcode. Barcode families containing $\geq$ 2 reads and have $\geq$ 90% reads being identical are selected. Sequences from selected barcode families were compared against annotated reference mutant sequences containing all possible single-nucleotide substitutions in the exon of interest, and the mutation in the reference mutant sequence was assigned to the matched barcode family. The frequency of the corresponding mutation was calculated by dividing the counts of the families containing the mutation by the total number of families.

## Generation of heatmaps

All heatmaps were generated using Morpheus (https://software.broadinstitute.org/morpheus). The mutation frequencies used in heatmaps in *Figure 3* and *Figure 3—figure supplement 1* were

corrected by the addition of $1 \times 10^{-5}$ (the detection limit at a barcode recovery of $1 \times 10^5$), $\log_{10}$ transformed and plotted.

## Statistics

The number of independent experiments and the statistical analysis used are indicated in the legends of each figure. Data are represented as mean ± SEM. p-Values were determined by Dunn's multiple comparison test following Kruskal–Wallis test, Holm–Sidak multiple comparisons test following one-way ANOVA, two-tailed Mann–Whitney U test, or two-sided Fisher's exact test. For correlation analysis, *Rho* and p values were derived from the Spearman correlation test. All statistical tests were performed using GraphPad Prism 6.

## Acknowledgements

We thank the head of and/or members of the laboratories of Drs. James Alvarez, Christopher Counter, David MacAlpine, and Nikoleta Tsvetanova (Duke University) for thoughtful discussions. This work was supported by the National Cancer Institute (R01CA94184 and P01CA203657 to CMC) and aided by core facilities supported by the Duke Cancer Institute (P30CA0124236).

## Additional information

### Funding

| Funder | Grant reference number | Author |
| --- | --- | --- |
| National Cancer Institute | R01CA94184 | Christopher M Counter |
| National Cancer Institute | P01CA203657 | Christopher M Counter |

The funders had no role in study design, data collection and interpretation, or the decision to submit the work for publication.

### Author contributions

Siqi Li, Conceptualization, Data curation, Formal analysis, Validation, Investigation, Visualization, Methodology, Writing - original draft, Writing - review and editing; Christopher M Counter, Conceptualization, Resources, Formal analysis, Supervision, Funding acquisition, Visualization, Writing - original draft, Project administration, Writing - review and editing

### Author ORCIDs

Christopher M Counter (iD) https://orcid.org/0000-0003-0748-3079

### Ethics

Animal experimentation: All animal experiments were approved by Duke IACUC.

### Decision letter and Author response

Decision letter https://doi.org/10.7554/eLife.67172.sa1
Author response https://doi.org/10.7554/eLife.67172.sa2

## Additional files

### Supplementary files

- Supplementary file 1. Information for mice and tumors involved in the tumorigenesis study.

- Supplementary file 2. Mutation frequency detected by maximum depth sequencing assay in the mutagenesis experiments.

- Supplementary file 3. Tumor sequencing, qPCR, and maximum depth sequencing assay primers.

- Transparent reporting form

## Data availability

All raw sequencing data has been deposited to NCBI Sequence Read Archive (SRA) under accession number PRJNA663179.

The following dataset was generated:

| Author(s) | Year | Dataset title | Dataset URL | Database and Identifier |
|---|---|---|---|---|
| Li S, Counter CM | 2021 | (1) Kras amplicon sequencing of urethane-induced mouse lung tumor. (2) Maximum depth sequencing of mouse lung tissue. | https://www.ncbi.nlm.nih.gov/bioproject/PRJNA663179 | NCBI BioProject, PRJNA663179 |

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
