## [Decision Letter]

**Acceptance summary:**

The question of why specific RAS mutations are selected in specific tumor types has long been a topic of discussion with important clinical implications. This work sheds important light on this topic, providing an intellectually satisfying and elegant demonstration of the way that KRAS mutation, mutagen, and gene copy interact in the setting of lung cancer. The work makes a valuable contribution to the field of RAS biology.

**Decision letter after peer review:**

Thank you for submitting your article "Signaling amplitude molds the Ras mutation tropism of urethane" for consideration by *eLife*. Your article has been reviewed by 3 peer reviewers, and the evaluation has been overseen by Erica Golemis as the Senior and Reviewing Editor. The following individuals involved in review of your submission have agreed to reveal their identity: Jonathan Chernoff (Reviewer #1); Ian A Prior (Reviewer #3).

Essential Revisions:

All reviewers felt that some writing changes would help with improving the clarity and accuracy of the work. Please address the following 3 critiques, from the 3 reviewers:

1) It is perhaps going too far to say in the Abstract and elsewhere that the transcriptional profiles reveal similar signaling amongst tumors driven by different mutations and Kras alleles. What they actually show is that a number of ERK-driven transcriptional events are roughly similar between the mutants. We don't know about the transcriptome in general nor about signaling in general beyond the ERK pathway. I think this point would be easy to address with some rewriting.

2) There is lack of cohesive language to describe the results. Please clarify the description

3) Transcription/translation mediated changes in Ras dosing to support tumourigenesis would be a very interesting observation if this was more conclusively demonstrated. Whilst I know that western blotting can't discriminate between mutant and wild type alleles – the data suggest an increase in total KRAS that should be detectable. Please provide either formal support for the Ras protein changes or rewriting to acknowledge the caveats in their interpretation.

Also, the comments relating to the ability of the models to fully understand what is happening do not require futher experiments but could be discussed. For example, genetic models that allow barcoding of tumours might help improve quantitation/analysis of life history/tumour specific signalling analysis in future iterations of this type of work.

*Reviewer #1 (Recommendations for the authors):*

I think the authors have provided strong in vivo evidence to support their theories about KRAS action. What is particularly appealing is that this work ties up many strands of "RAS-ology," such as the inclusion of rare codons in KRAS, the reasons for mutational tropism (e.g., Q61 vs G12/13 mutations), and a cogent rationale for the existence of allellc imbalance in mutant KRAS expression.

*Reviewer #3 (Recommendations for the authors):*

Is amplitude the right word? Target gene expression does not necessarily allow this specific feature of signalling kinetics to be determined.

---

## [Author Response]

Essential Revisions:All reviewers felt that some writing changes would help with improving the clarity and accuracy of the work. Please address the following 3 critiques, from the 3 reviewers:1) It is perhaps going too far to say in the Abstract and elsewhere that the transcriptional profiles reveal similar signaling amongst tumors driven by different mutations and Kras alleles. What they actually show is that a number of ERK-driven transcriptional events are roughly similar between the mutants. We don't know about the transcriptome in general nor about signaling in general beyond the ERK pathway. I think this point would be easy to address with some rewriting.

As requested, we revised this statement everywhere it occurred to instead state the actual data, and also noted the caveats to interpreting this type of analysis (also see our response to essential revision #3).

2) There is lack of cohesive language to describe the results. Please clarify the description

As requested, we revised the entire manuscript for clarity and cohesiveness, including revising the concluding sentences in each section of the discussion, and referring to alleles, results, assays, concepts and so forth in a common way.

3) Transcription/translation mediated changes in Ras dosing to support tumourigenesis would be a very interesting observation if this was more conclusively demonstrated. Whilst I know that western blotting can't discriminate between mutant and wild type alleles – the data suggest an increase in total KRAS that should be detectable. Please provide either formal support for the Ras protein changes or rewriting to acknowledge the caveats in their interpretation.

As requested, we revised the text to note these caveats and also better framed this data with respect to our conclusions. We also cite that the *Kras^ex3op^* allele is expressed two-fold over the native allele (*see* Figure 1h in Pershing et al., J Clin Invest, 2015).

Also, the comments relating to the ability of the models to fully understand what is happening do not require futher experiments but could be discussed. For example, genetic models that allow barcoding of tumours might help improve quantitation/analysis of life history/tumour specific signalling analysis in future iterations of this type of work.

As requested, we added these concepts to the discussion.

Reviewer #3 (Recommendations for the authors):Is amplitude the right word? Target gene expression does not necessarily allow this specific feature of signalling kinetics to be determined.

As requested, we removed the word amplitude from the title and throughout the text.